# Recombinant Photolyase-Thymine Alleviated UVB-Induced Photodamage in Mice by Repairing CPD Photoproducts and Ameliorating Oxidative Stress

**DOI:** 10.3390/antiox11122312

**Published:** 2022-11-22

**Authors:** Zhaoyang Wang, Ziyi Li, Yaling Lei, Yuan Liu, Yuqing Feng, Derong Chen, Siying Ma, Ziyan Xiao, Meirong Hu, Jingxian Deng, Yuxin Wang, Qihao Zhang, Yadong Huang, Yan Yang

**Affiliations:** 1Department of Cell Biology, Jinan University, Guangzhou 510632, China; 2TYRAN Cosmetics Innovation Research Institute, Jinan University, Guangzhou 511447, China; 3Department of Pharmacology, Jinan University, Guangzhou 510632, China; 4Guangdong Province Key Laboratory of Bioengineering Medicine, Guangzhou 510632, China

**Keywords:** recombinant photolyase, photorepair, DNA damage, oxidative stress

## Abstract

Cyclobutane pyrimidine dimers (CPDs) are the main mutagenic DNA photoproducts caused by ultraviolet B (UVB) radiation and represent the major cause of photoaging and skin carcinogenesis. CPD photolyase can efficiently and rapidly repair CPD products. Therefore, they are candidates for the prevention of photodamage. However, these photolyases are not present in placental mammals. In this study, we produced a recombinant photolyase-thymine (rPHO) from *Thermus thermophilus* (*T. thermophilus*). The rPHO displayed CPD photorepair activity. It prevented UVB-induced DNA damage by repairing CPD photoproducts to pyrimidine monomers. Furthermore, it inhibited UVB-induced ROS production, lipid peroxidation, inflammatory responses, and apoptosis. UVB-induced wrinkle formation, epidermal hyperplasia, and collagen degradation in mice skin was significantly inhibited when the photolyase was applied topically to the skin. These results demonstrated that rPHO has promising protective effects against UVB-induced photodamage and may contribute to the development of anti-UVB skin photodamage drugs and cosmetic products.

## 1. Introduction

Ultraviolet (UV) radiation has direct and long-term harmful effects on the skin, resulting in acute erythema (sunburn), collagen and elastin degradation, and wrinkles (photoaging) [1]. There are three different wavelength ranges of UV light: UVA (320–400 nm), UVB (280–320 nm), and UVC (200–280 nm) [2,3]. Photochemical cellular damage occurs when UVB radiation penetrates the epidermis and upper dermis layers. UVB can be directly absorbed by DNA and induces mutagenic DNA photoproducts [4,5,6]. Pyrimidine-pyrimidone (6–4) photoproducts ((6–4) PPs) and cyclobutane pyrimidine dimers (CPDs, 75%) are the main products of UVB-induced DNA damage [7,8]. CPDs and (6–4) PPs accumulate and inhibit the transcription and replication of DNA, eventually disrupting cell function. UVB also causes leukocyte infiltration and increases the production of reactive oxygen species (ROS) [9], which damage DNA. Excessive ROS production affects protein stability (e.g., by increasing collagen fragmentation), which may lead to oxidative stress and inflammation [10,11]. If these lesions are not removed in a timely manner, they will result in cell death.

Numerous DNA repair mechanisms have evolved to prevent the adverse effects of UVB damage on DNA, including nucleotide excision repair (NER) and photoreactivation enzymes/photolyases. DNA photolyases are considered to be the oldest DNA repair enzymes [12]; they contain the catalytic cofactor flavin adenine dinucleotide (FAD) and a photoantenna. They are responsible for repairing DNA damage in a light-dependent manner [13] by recognizing and binding to DNA lesions, such as CPDs or (6–4) PPs. FAD is essential for the DNA repair activity of photolyases. Photoreactivation is an efficient DNA repair process catalyzed by photolyases, resulting in the removal of CPDs or (6–4) PPs from DNA within an hour [14]. The reaction is divided into two main stages. First, the U-shaped structural domain FADH of the DNA photolyase complements the damaged site, thus firmly binding to the CPD and forming a stable enzyme–substrate complex that further exposes the damaged site in the double-helix structure. This stage is independent of light [15,16]. After the antenna molecule absorbs a photon of UVA/blue light, it transfers the excitation energy to the FADH through Förster dipole–dipole resonance; then, the electron is transferred to the CPD or (6–4) PP, repairing the DNA [13,14,17].

Since actinic keratosis and skin cancer could be prevented and treated with photolyases, it is becoming increasingly important to screen for new photolyases and study their biochemical properties. Several bacteria, fungi, and eukaryotes were found to contain photolyases [18,19,20]. A sunscreen extract containing DNA photolyase from *Anacystis nidulans* exhibited photoprotective effects [21]. It is worth noting that extracts isolated from bacteria may contain lipopolysaccharides, which can trigger allergic reactions or skin diseases [22]. However, affinity-chromatography-purified recombinant photolyase exhibits high reactivity and no impurities [23,24].

Here, we produced a recombinant photolyase-thymine protein (rPHO) from *T. thermophilus.* The rPHO belongs to the DNA photolyase class-1 family, which is involved in the repair of UV-radiation-induced DNA damage and catalyzes the light-dependent monomerization (300–600 nm) of CPDs, which are formed between adjacent bases on the same DNA strand upon exposure to ultraviolet radiation. The amino acid sequence of rPHO has two different protein domains: (i) an antenna domain that holds the chromophore, and which transfers energy by resonance to the FAD; and (ii) the photolyase domain that binds the FAD (responsible for removing the UV-induced DNA lesion). The FAD is noncovalently attached at the active site. After light absorption, rPHO repairs the pyrimidine dimers to their monomeric forms by transferring electrons to the dimers [25]. We characterized the photorepair activity of rPHO in vitro and in vivo. The results suggested that rPHO is a potential agent for alleviating UVB-induced photodamage.

## 2. Materials and Methods

### 2.1. Recombinant Photolyase Expression and Purification

A photolyase gene (*Phr*) fragment from *T. thermophilus* (strain ATCC BAA-163/DSM7039/HB27) was synthesized and supplied by GenScript China Inc. (Nanjing, China). The synthesized photolyase gene fragment was inserted into the pET-20b after transformation into *Escherichia coli* BL21(DE3) and sequencing (Sangon, Shanghai, China). To induce protein expression, bacteria were grown in LB (Amp+) flasks at 37 °C to an OD of 0.6. Thereafter, cells were collected and broken up. Then, cells were centrifuged for 15 min. The supernatant and precipitate were analyzed by SDS-PAGE. The rPHO contained His-tag. Ni-NTA was used to purify the recombinant photolyase. Clarified bacterial cell lysates were applied to a Ni-NTA FPLC column or to a manually prepared Ni-NTA column. The unbound protein fraction was washed off with lysis buffer containing 20 mM imidazole, and the elution of proteins specifically bound to the FPLC column was carried out with a 20–500 mM imidazole gradient. Proteins were also eluted in bulk from Ni-NTA manually prepared columns with a buffer containing 250 mM imidazole. Then, an endotoxin removal agarose resin kit (YEASEN, 20518ES10) was used to remove endotoxin according to the manufacturer’s manual. In brief, the endotoxin removal agarose resin was fully mixed, an appropriate amount of grout was absorbed and added to the appropriate chromatographic column with the pyrogenless gun head, and bubbles were avoided. The lower outlet was opened to remove the protective liquid, and 3 mL regenerated liquid was used for cleaning. The flow rate was controlled at 0.25 mL/min, or <10 drops/min. We repeated the procedure at least two times to ensure that there were no endotoxins in the column. Then, 3 mL of regenerated liquid was used for cleaning, and the flow rate was controlled at 0.25 mL/min to remove endotoxins. The sample was added to the balanced resin, and the flow rate was adjusted to 0.25 mL/min, or <10 drops/min. When about 1 mL of outflow liquid had drained, it was collected, and 1 mL of the equilibrium liquid was added to continue the collection.

### 2.2. Enzyme Activity Assay of CPD Photolyase rPHO In Vitro

Oligo (dT)16 (5′-TTTTTTTTTTTTTTTT-3′) were purchased from Sangon (Shanghai, China). We prepared CPD oligonucleotides using a UVB lamp (SIGMA, SH2 B-J, 10 mW/cm^2^), irradiating them until their OD260 no longer decreased. The enzyme activity was measured as previously described [26]. The assay system contained 50 μg/mL of rPHO and 0.2 μM UV-oligo (dT)16. The mixtures were incubated for 60 min under a daylight lamp (Philipp 8W, Shanghai, China). Then, the solution was boiled for 5 min and centrifuged. The detection wavelength was 260 nm.

### 2.3. Cellular Uptake Assay

To explore the uptake of rPHO by HaCat cells, FITC was used to label the rPHO (FITC-rPHO), and then HaCaT cells were treated with different concentrations of FITC-rPHO at 37 °C, 5% CO_2_ for 4 h, while the control group was not subjected to any treatment. After that, the cells were fixed for 10 min with 4% paraformaldehyde and washed three times with PBS. Finally, the nuclei were stained with DAPI at room temperature for 10 min, and images were taken with an LSM laser confocal microscope.

### 2.4. Skin Permeation Assay

To explore the distribution of rPHO in the skin tissue of mice, FITC was used to label rPHO (FITC-rPHO gel), and 1% carbomer was added to make a gel preparation. The back-hair removal of 6-week-old female Kunming mice was performed on the second day by irradiation from the ultraviolet lamp for 2 h. FITC-rPHO gel at different concentrations was applied to the back of the mice, and the control group was administered unlabeled FITC-rPHO. The mice were kept for 24 h and then euthanized. The back skin tissues were taken and embedded into Tissue-Tek OCT at −20 °C. The embedded skin was then cut into sections with a thickness of 5μm using a cryotome (CryoStar NX50, Thermo Fisher, Waltham, MA, USA). Finally, the nuclei were stained using DAPI, and the images were taken using an LSM laser confocal microscope. The skin delivery of the FITC-rPHO gel was visualized by assessing the penetration of the fluorescent dyes.

### 2.5. Cytotoxicity Assay

HaCaT cells were treated with photolyase for 2 h. Then, these cells were exposed to a UVB lamp (SIGMA, SH2 B-J, 10 mW/cm^2^, Shanghai, China) for 2 min 15 s at a dose of 2 mJ/cm^2^, following a previously described protocol [27], and cultured for an additional 12 h. Finally, CCK8 (10 μL/well) was applied for 1 h and measured at 490 nm.

### 2.6. Detection of Apoptosis

HaCaT were pretreated with photolyase for 2 h. Then, cells were UVB-irradiated for 2 min and 15 s (2 mJ/cm^2^) and cultured for 24 h. The adhering cells were collected by centrifugation. The cells were resuspended in annexin V-EGFP binding solution (Beyotime, Shanghai, China). Annexin V-EGFP and PI were added, and the mixture was left for 15 min. A flow cytometer was used to detect cell apoptosis.

### 2.7. Laser Confocal Analysis of DNA Contents

Cells were fixed with 4% paraformaldehyde (PFA) for 20 min, blocked for 1 h, incubated with γ-H2AX (Beyotime, Beijing, China), and then incubated with Alexa Fluor 488. Nuclei were stained with DAPI. Green FITC indicated DNA, and blue DAPI staining indicated nuclei in the laser confocal pictures.

### 2.8. ROS Determination

A Reactive Oxygen Species Assay Kit (Beyotime Institute of Biotechnology, Wuhan, China) was used to detect intracellular ROS, following the manufacturer’s protocol. Dichlorodihydrofluorescein diacetate (DCFHDA) was added, and the mixture was incubated for 20 min. ROS production was measured using ImageJ software and observed using an inverted confocal laser microscope. These experiments were repeated sequentially three times.

### 2.9. Quantitative Real-Time PCR (qRT-PCR)

The mRNA expression levels of genes were analyzed using the CFX96 real-time PCR system (Bio-Rad, Hercules, CA, USA). The primer sequences (Table 1) are presented below.

### 2.10. Western Blot Assay

Photolyase was applied to the cells for 2 h, followed by UVB irradiation for 2 min and 15 s and culturing for another 6 h. Cell lysates were collected using RIPA lysates (Beyotime, Shanghai, China). Primary antibodies were incubated overnight at 4 °C, followed by secondary antibodies for 1 h at room temperature. The following antibodies were used: BCL-2 monoclonal antibody (ThermoFisher Scientific, Cambridge, MA, USA); Bax monoclonal antibody (ThermoFisher Scientific; USA33-6400); caspase 3 monoclonal antibody (ThermoFisher Scientific, 700182); and β-Actin monoclonal antibody (ThermoFisher Scientific, MA1-140).

### 2.11. Animal Experiment

Kunming mice were purchased from the Experimental Animal Center of Guangdong Province, China. All animal experiments were conducted according to the care and use of animals guideline and approved by the Institutional Animal Care and Use Committee of Jinan University (approval number: IACUC; issue No: 20210301-14). The process of fabricating the gel formulations, in brief, was as follows: rPHO at different concentrations (2.5 g, 5 g, and 10 g) was dissolved in 100 mL PBS, and then 1 g of carbomer was added and stirred until it was completely dissolved, forming a gel.

Six-week-old female mice were randomly divided into seven groups (7 mice per group). One week later, the hair on the back of the mice was removed using hair-removal cream (ICE King), and the next day, mice were exposed to UVB (16.6 mW/cm^2^) for 2 h every day; it was calculated that the mice received 120 mJ/cm^2^ of UVB irradiation per day. The irradiation dose was calculated using the following formula: dose (mJ/cm^2^) = exposure time (sec.) × intensity (mW/cm^2^) [28].

The mice were placed in a relatively airtight box with plenty of food and water; then, an ultraviolet lamp (Philipp TUV8W) was placed at the upper end of the box, and the mice were irradiated for 2 h every day for 15 days. The blank group was not subjected to irradiation. Then, the photolyase gel formulations of different concentrations were applied to the backs of the mice. The PBS group was treated with gel formulations without rPHO. The dose was administered continuously for 15 days, and UVB irradiation continued for 2 h daily during the dosing regimen (irradiation prior to application of the drug). After 15 days, images of the mouse skin were obtained, and blood and skin tissues of the mice were collected. The skin tissues were fixed with 4% PFA, embedded, and sliced. The expression levels were measured by qRT-PCR. The enzymatic activity was detected using the following kits: SOD Activity Detection Kit (Beyotime, Shanghai, China; S0101M); Glutathione Peroxidase (GSH-PX) ELISA Kit (Signalway Antibody, Greenbelt, MD, USA, EK6277); Hydroxyproline (HYP) Content Detection Kit (Solarbio, Beijing, China; BC0255); and MDA Content Detection Kit (Solarbio; BC025).

### 2.12. Statistical Analysis

Data are presented as means ± SD. Data were analyzed by one-way ANOVA (analysis of variance) or Student’s *t*-tests, as appropriate. *p* value < 0.05 was considered statistically significant.

## 3. Results

### 3.1. Recombinant Photolyase-Thymine Protein (rPHO) from T. thermophilus Exhibited CPD Photorepair Activity

Photolyase-thymine from *T. thermophilus*, also named deoxyribodipyrimidine photolyase, belongs to the DNA photolyase class-1 family. It is able to repair UVB-induced DNA damage using electron transfer (Figure 1A). Here, we synthesized the open reading frame (ORF) of the photolyase-thymine gene (*Phr*) fragment, which contained 1248 bases encoding 416 amino acids. The ORF of *Phr* was cloned and expressed in *E. coli* BL21(DE3). The amino acid sequence of *Phr* had a photolyase domain, which binds to the FAD (Figure 1B). The recombinant photolyase-thymine protein (rPHO) had a molecular weight of 47 kDa, as determined by SDS-PAGE (Figure 1C). The molecular mass and pI, as predicted using the ExPASy server, were 47.9 kDa and 9.26, respectively.

An in vitro enzymatic activity assay was used to determine the photorepair activity of photolyase. Briefly, CPD photoproducts were produced when Oligo (dT)16 was exposed to UVB. Our results showed that the value of Oligo (dT)16 remained stable at about 0.26 after irradiation by UVB, indicating the formation of CPD photoproducts. During photolyase repair, the number of pyrimidine monomers gradually increased, resulting in an increase in absorbance at 260 nm. rPHO increased the value to 0.35 in the presence of light. These data suggested that rPHO exhibited photorepair activity, and that CPD photoproducts could be repaired to pyrimidine monomers (Figure 1D).

### 3.2. rPHO Was Taken Up by HaCaT Cells

The analysis of rPHO uptake capacity and skin permeability was an important first step before studying the function of rPHO in repairing UV damage. The rPHO uptake capacity in cells and mouse skin was measured. The fluorescein isothiocyanate (FITC)-labeled rPHO were used to examine the ability of HaCat to uptake rPHO. We found that the incubation of HaCaT cells with FITC-labeled rPHO for 4 h resulted in robust uptake responses by the FITC-labeled rPHO. Furthermore, we observed green fluorescence in the cytoplasm and nucleus, which indicated that the rPHO was taken up into cells and was localized to the nucleus and cytoplasm (Figure 2A). Additionally, to visualize the rPHO penetration pathway in the mouse skin, FITC-labeled rPHO carbomer gels were applied to the dorsal skin for 24 h, and the fluorescence was observed in the keratinocytes of the skin epidermis (Figure 2B). The results suggested that rPHO was taken up into the cells and absorbed by the keratinocytes of the skin epidermis.

### 3.3. rPHO Protected HaCaT Cells against UVB-Induced Reduction in Cell Viability

HaCaT cells were used to examine the protective effect of rPHO on UVB-induced cellular damage. Cells were incubated with different doses of rPHO for 2 h prior to UVB exposure (2 mJ/cm^2^) (Figure 3A). HaCaT cells tended to become rounded, shrunk, and detached and eventually underwent apoptosis after UVB irradiation; these changes in cell morphology were attenuated by rPHO treatment (Figure 3B). The number of viable cells decreased by 41% after UVB exposure. rPHO increased cell viability to 79.2% and 83.9% at 12.5 and 25 μg/mL, respectively, and to approximately 100% at 50 μg/mL. (Figure 3C). Then, a calcein-AM and propidium iodide (calcein-AM/PI) assay was used to visualize living (green) and dead cells (red), respectively. The number of PI-positive cells (red) increased dramatically after UVB exposure, indicating that rPHO treatment could efficiently protect HaCaT cells against UVB-induced cell death at 12.5, 25, and 50 μg/mL (Figure 3D). These data suggested that rPHO could suppress UVB-induced cell death.

### 3.4. rPHO Prevented UVB-Induced Photoaging in Mice

We then examined the anti-photoaging effects of rPHO in vivo. Mice were exposed to repetitive UVB (120 mJ/cm^2^) radiation for 15 days. Next, PBS or rPHO at concentrations of 2.5%, 5%, and 10% were applied to the dorsal skin for 15 days before irradiation (Figure 4A). The untreated skin was used as a normal control. UVB-irradiated mice with no protection were used as a model group. After 15 days of treatment, we observed erythema and wrinkles on the dorsal skin of the UVB-irradiated (model group) and PBS-treated mice. The erythema and wrinkle formation were attenuated by rPHO treatment at 2.5%, 5%, and 10% (Figure 4B). The thickness of the epidermal layer increased from 0.98 μm in normal skin to 3.79 μm and 4.81 μm in the model and PBS group, respectively. In contrast, rPHO significantly inhibited UVB-induced epidermal thickening. The epidermal thickness (1.20 μm) in the 10% photolyase treatment group was similar to that of the normal control group (0.98 μm) (Figure 4C,D), indicating that rPHO could suppress UVB-induced epidermal hyperplasia and protect against macroscopic skin damage induced by UVB irradiation.

### 3.5. rPHO Inhibited the UVB-Induced Degradation of Collagen In Vivo

The arrangement and distribution of collagens are essential for maintaining skin structure. The UVB-induced degradation of collagen causes wrinkling and laxity [29]. Thus, Masson’s trichrome and picrosirius red staining were used to evaluate the deposition and distribution of collagen in the skin after rPHO treatment. Compared to normal skin, UVB irritation led to reduced collagen deposition and irregularly distributed collagen fibers in the model and PBS-treated groups. After rPHO treatment, the density of collagen fibers increased, and the arrangement of collagen fibers was more regular and orderly than that in the model and PBS-treated groups (Figure 5A,B). Moreover, the collagen I/III ratio was significantly lower in the model and PBS-treated groups than in normal skin. rPHO significantly elevated the ratio of collagen I/III. The ratio in skin treated with rPHO (at 2.5%, 5%, and 10%) was similar to that in normal skin (Figure 5C). Finally, the collagen content in the skin was determined and quantified by a hydroxyproline (HYP) assay. UVB induced a significant reduction in the collagen content, while rPHO increased the collagen content by increasing the hydroxylation of collagen. The HYP level decreased from 32.68 μg/mL to 21.51 μg/mL after UVB irradiation and increased from 21.51 μg/mL to 27.17 μg/mL after treatment with rPHO (10%) (Figure 5D).

These results demonstrated that rPHO improved the collagen fiber arrangement and protected collagen in the skin from UVB-induced degradation, providing an effective approach for maintaining a topographically smooth skin structure and reducing wrinkles after UVB exposure.

### 3.6. rPHO Attenuated UVB-Induced Oxidative Stress and Inflammatory Responses in Mice

We also evaluated UVB-induced tissue damage in the dorsal skin by measuring malondialdehyde (*MDA*), a product of lipid peroxidation, which may contribute to oxidative stress. UVB irradiation caused a significant increase in MDA from 5.07 nM in normal skin to 10.40 nM and 9.99 nM in the model and PBS groups, respectively. rPHO treatment prevented the progressive increase in MDA after UVB exposure. The levels of MDA were comparable to those in normal skin after rPHO treatment (10%) (Figure 6A). Since MDA contributes to oxidative stress, the activity levels of antioxidant enzymes (SOD and GSH-Px) were determined. The SOD and GSH-Px levels in dorsal skin were significantly downregulated following UVB irradiation. However, rPHO treatment (10%) significantly increased the activity of SOD and GSH-Px to levels similar to those in the normal control group (Figure 6B,C). These results indicated that rPHO could attenuate UVB-induced oxidative stress.

Additionally, UVB exposure increases the levels of pro-inflammatory cytokines, such as IL-6, IL-1β, and TNF-α, in turn leading to an inflammatory disequilibrium in the skin [30]. To determine the effect of rPHO on UVB-induced cutaneous necro-inflammation, we examined IL-6, IL-1β, and TNF-α expression in the dorsal skin. The expression levels of IL-6, IL-1β, and TNF-α increased significantly after UV exposure, while rPHO treatment significantly decreased IL-6, IL-1β, and TNF-α expression (Figure 6D–F). Since oxidative stress and inflammatory stress are involved in regulating UVB-induced cell apoptosis [31], we then examined the levels of anti-apoptotic and pro-apoptotic proteins. The expression levels of pro-apoptotic proteins Bax and caspase 3 were significantly upregulated after UVB exposure, and the anti-apoptotic protein Bcl-2 expression was downregulated. However, rPHO significantly inhibited Bax and caspase 3 expression and upregulated Bcl-2 expression, thereby increasing the ratio of Bcl-2/Bax (Figure 6G,H). Taken together, these results demonstrated that rPHO inhibited UVB-induced oxidative stress, inflammatory responses, and apoptosis in the skin.

### 3.7. rPHO Reduced UVB-Induced DNA Damage in HaCaT Cells

In order to understand how rPHO protects against photodamage induced by UVB, HaCaT cells were treated with rPHO for 24 h after UVB irradiation (120 mJ/cm^2^). The γ-H2AX formation, a DNA double-strand break marker, was assessed by immunofluorescence staining. A high percentage of γ-H2AX-positive cells was detected after UVB exposure. However, after treatment with rPHO, the number of γ-H2AX-positive cells decreased significantly. rPHO at 50 μg/mL could efficiently prevent DNA breaks (Figure 7A). Then, the formation of comet tails was used to assess UVB-induced DNA fragmentation. UVB exposure significantly increased the average percentage of tail DNA. As a result of rPHO treatment, the tail DNA percentage decreased significantly, and the effect of rPHO was dose-dependent (Figure 7B). We further measured the length of the comet tail. The longest comet tail was detected in HaCaT cells irradiated with UVB. The tail length was shortened by rPHO. HaCaT cells treated with 50 μg/mL of rPHO exhibited intact nuclear DNA after UVB exposure, similar to normal HaCaT cells (Figure 7C,D). These results demonstrated that DNA damage caused by UVB could be effectively prevented by rPHO, which exerted a protective effect on DNA.

### 3.8. rPHO Exhibited ROS-Scavenging and Anti-Inflammatory Activity

To further explore the ROS-scavenging ability of rPHO, we performed an ROS scavenging assay by DCFH-DA on HaCaT cells. UVB-irradiation (2 mJ/m^2^) induced ROS generation in HaCaT cells. In cells treated with rPHO, there was a clear decrease in ROS production (Figure 8A). Then, the ROS levels in HaCaT cells were measured by flow cytometry. ROS levels were significantly higher in UVB-irradiated cells than in normal cells. However, the increase in ROS levels in irradiated cells was significantly attenuated by treatment with rPHO at concentrations of 12.5, 25, and 50 µg/mL (Figure 8B,C). Since the production of ROS might cause an inflammatory response, we examined the mRNA expression of IL-6, IL-1β, and TNF-α in cells. The IL-6, IL-1β, and TNF-α expression levels were significantly upregulated by UVB, while rPHO inhibited these UVB-induced increases in inflammatory factor expression (Figure 8D–F). Together, these data confirmed that rPHO could effectively reduce ROS production and inflammatory responses.

### 3.9. Photolyase Inhibited UVB-Induced Apoptosis in HaCaT Cells

Cytochrome C release is required for caspase activation and apoptosis induction [32]; therefore, we examined the cytoplasmic cytochrome c levels by immunofluorescence staining. UVB exposure induced significant cytochrome C accumulation in the cytoplasm, while rPHO treatment significant suppressed cytochrome c release (Figure 9A). Cytochrome c release is influenced by the relative expression levels of Bcl-2, Bax, and caspase 3. We found that Bcl-2 levels were significantly lower in UVB-irradiated HaCaT cells than in control cells, while caspase 3 and Bax expression levels were significantly increased by UVB irradiation (Figure 9B–D). Caspase 3 expression was significantly inhibited in HaCaT cells treated with rPHO. The ratio of Bcl-2/Bax was also elevated by rPHO. Moreover, UVB irradiation resulted in a significantly higher rate of apoptosis (18.17%) than that in non-irradiated cells (3.62%), whereas the rate of apoptosis was only 3.50% in the rPHO group (Figure 9E,F). These findings suggested that rPHO protects HaCaT cells against UVB-induced apoptosis by modulating cytochrome c release and apoptosis-related protein expression.

## 4. Discussion

Sunscreens containing antioxidants contribute to the repair of damaged DNA and improve resistance to acute photodamage [33,34,35]. However, there may be some side effects associated with the long-term excessive use of antioxidants with low activity and poor stability [36]. Therefore, the development of new agents to prevent and treat skin photoaging is of great urgency. Photolyases have the potential to be used to prevent and treat actinic keratosis and skin photoaging. However, placental mammals do not have their own DNA photolyases [37]. Accordingly, an extract of photolyase from archaea or bacteria has been used for sunscreens [38,39,40]. Photolyases isolated from archaea or bacteria often contain lipopolysaccharides and bacterial impurities, which may induce allergic reactions or skin diseases [29]. However, the isolation of photolyases from their natural hosts is usually difficult and inefficient.

In this study, the photolyase gene (*Phr*) from *T. thermophilus* was synthesized and expressed in *E. coli*, encoding a protein (rPHO) that had CPD repair activity. This avoided bacterial lipopolysaccharide contamination, providing an alternative strategy for the extraction of photolyase from *T. thermophilus*. The recombinant production of proteins in *E. coli* is beneficial for structural and functional analyses. Furthermore, *E. coli* also exhibits quick growth, high biomass generation, and low production costs, promoting its application on an industrial scale. More importantly, rPHO can be used to investigate the therapeutic potential and mechanism of action of *T. thermophilus* photolyase.

Though the photolyase gene from *T. thermophilus* has previously been cloned and sequenced [41,42], the function of the recombinant photolyase protein and its clinical value and mechanisms for skin photodamage are unclear. We found that UVB irradiation caused wrinkles and blotches on the skin of mice, whereas after 15 days of treatment with rPHO, the skin became smooth, indicating that rPHO could protect against UVB-induced skin damage. We evaluated the mechanisms involved in the protective effects of rPHO, revealing that rPHO reduced UVB-induced DNA damage. Furthermore, rPHO reduced ROS and MDA levels and enhanced the activity of SOD, CAT, and GSH-Px to protect against UVB-induced oxidative stress. Additionally, our results indicated that rPHO treatment significantly inhibited the expression of TNF-α, IL-6, and IL-β in HaCaT cells. These results indicated that rPHO significantly inhibited UVB-induced oxidative stress, inflammatory responses, and cell apoptosis. However, it would be useful to conduct a comparison with photolyase. Unfortunately, there was no standard available for photolyase. Therefore, further studies are needed to confirm the precise function of rPHO in photodamage by comparison with a photolyase standard.

Additionally, the transdermal delivery of proteins is always challenging due to their poor transdermal permeability. Determining the rPHO uptake capacity and skin permeability was an important first step before studying the function of rPHO in repairing UV damage. Paola Hernandez et al. reported recombinant CPD-photolyase PhrAHym from the Antarctic bacterium *Hymenobacter sp. UV11*. The PhrAHym repaired the CPD photoproducts in a highly efficient manner and reduced γH2AX formation. These results are similar to those of our study [43]. However, the cell transfection with PhrAHym was achieved using Lipofectamine LTX (Invitrogen) transfection mix. This transfection mix cannot be used in real-world applications. In our study, we demonstrated that rPHO could be taken up by HaCaT cells and keratinocytes.

Since transdermal drug delivery systems include transdermal patches, gels, and emulgels, in this study, to enhance the transdermal permeation of rPHO, we prepared an rPHO carbomer gel. The presence of the carbomer ensured optimal hydration for the skin [44] and promoted the effective absorption of rPHO. Furthermore, photodamage often involves impaired skin barrier function [45], which facilitated the passage of rPHO through the skin barrier. These factors contributed to enhancing the transdermal permeation of rPHO. However, it is worth noting that the transdermal absorption of rPHO may be limited. Further explorations are warranted regarding the transdermal and topical formulation application of rPHO.

Nevertheless, we demonstrated that rPHO accelerated the repair of UVB-induced DNA damage by repairing CPD photoproducts; inhibited UVB-induced oxidative stress; and eliminated ROS production, further inhibiting inflammatory responses and cell apoptosis. Our results supported the potential application of rPHO in protection against UVB-induced photodamage, as well as in the development of drugs to prevent UVB-induced skin damage.

## 5. Conclusions

Recombinant photolyase-thymine protein (rPHO) exhibited photoreactivation activity and repaired CPD photoproducts into monomers. rPHO was taken up into cells and was localized to the nucleus and cytoplasm in vitro. In the animal experiments, rPHO carbomer gel resulted ensured optimal hydration for the skin and promoted the effective absorption of rPHO. When the rPHO gel was applied topically to the skin, it increased the survival of UVB-exposed keratinocytes and prevented photodamage in mice skin by repairing DNA breaks, ameliorating oxidative stress, and regulating inflammatory factors, thereby inhibiting the degradation of collagen and apoptosis (Figure 10). More importantly, recombinant DNA technology provides an economical means for rPHO production and is an effective alternative to extraction from *T. thermophilus* for pharmaceutical and cosmetic applications. These findings demonstrate the promising role of rPHO in attenuating UVB-induced photoaging and inflammatory responses in the skin; rPHO may contribute to the development of pharmaceutical and cosmetic products with potential activity against UVB-induced photoaging.

## Figures and Tables

**Figure 1 antioxidants-11-02312-f001:**
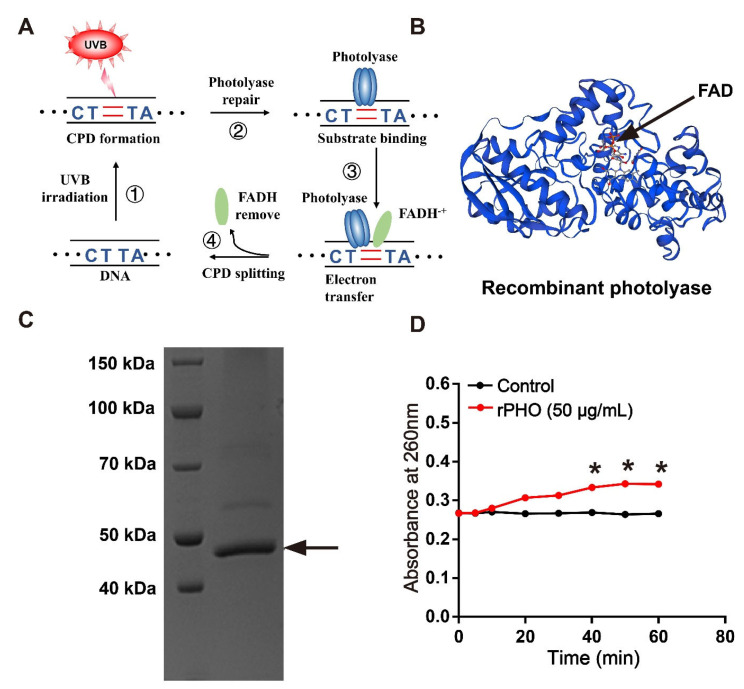
Recombinant photolyase-thymine protein (rPHO) from *T. thermophilus* displayed CPD photorepair activity. (**A**) Mechanism underlying photolyase-mediated repair of UVB-induced DNA damage. (**B**) Structure of recombinant photolyase-thymine (rPHO). (**C**) The purified protein was separated by SDS-PAGE; arrow shows the target protein (rPHO). (**D**) Enzymatic activity assay was used to examine the photorepair activity of rPHO. CPD photoproducts were formed by exposure of Oligo (dT)16 to UVB; then, the CPD photoproducts were incubated with rPHO (50 μg/mL) for 0, 5, 10, and 20 min, followed by the detection of the OD at 260 nm. * *p* < 0.05.

**Figure 2 antioxidants-11-02312-f002:**
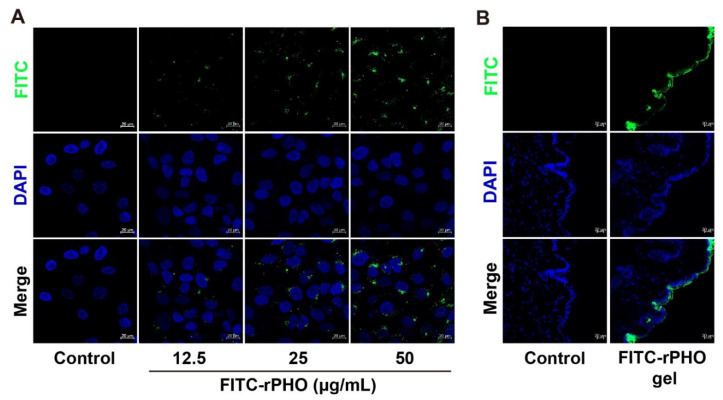
rPHO was taken up into cells and absorbed by the keratinocytes of the skin epidermis. (**A**) HaCaT cells were treated with different concentrations of rPHO-FITC. (**B**) FITC-labeled rPHO carbomer gels were applied to the dorsal skin for 24 h. Vertical skin sections with a thickness of 10 µm were extracted with cryotome and observed under a fluorescence microscope for skin-associated fluorescence. Scale bar = 20 µm.

**Figure 3 antioxidants-11-02312-f003:**
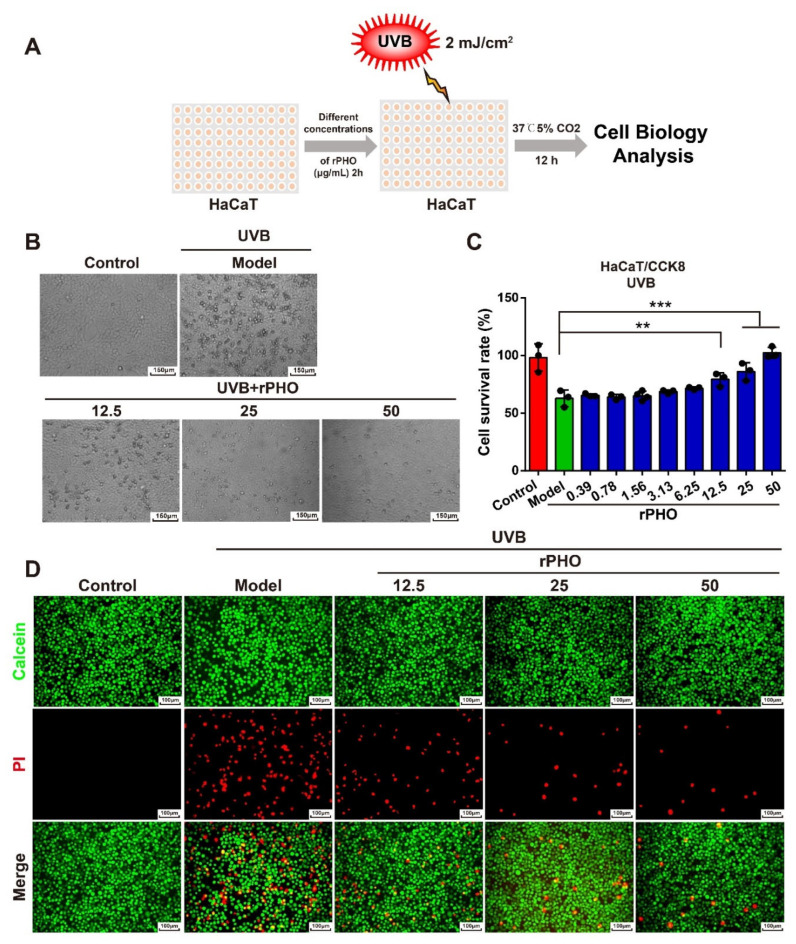
rPHO significantly reduced UVB-induced death of HaCaT cells. (**A**) UVB irradiation experimental design. (**B**) Images of cells were obtained under a bright field by a microscope after rPHO treatment. Scale bar = 150 μm. (**C**) Cell survival was measured. (**D**) Cells were stained with calcein-AM/PI; dead cells were stained red and live cells were stained green. Scale bar = 100 μm. Data are presented as means ± SD (*n* = 3). ** *p* < 0.01, *** *p* < 0.001.

**Figure 4 antioxidants-11-02312-f004:**
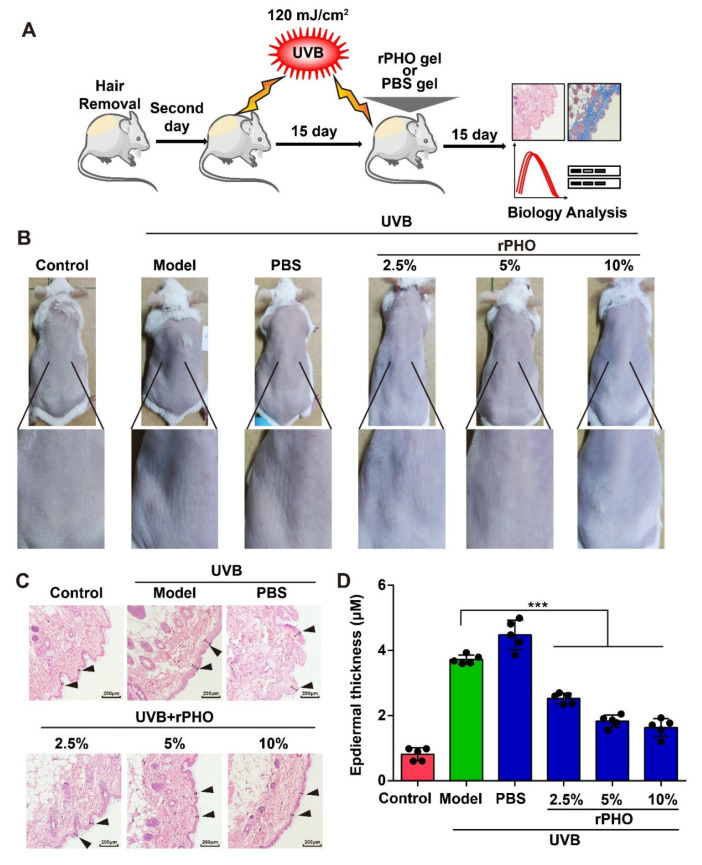
rPHO prevented UVB-induced photoaging in mice. (**A**) Schematic diagram of UVB irradiation and rPHO administration. (**B**) Representative images of the morphological changes of the dorsal skin in each group after rPHO treatment for 15 days. (**C**) Representative images of HE staining after rPHO treatment for 15 days. Scale bar = 100 μm. (**D**) The thickness of the epidermis was measured using ImageJ. Data are presented as means ± SD (*n* = 5). *** *p* < 0.001.

**Figure 5 antioxidants-11-02312-f005:**
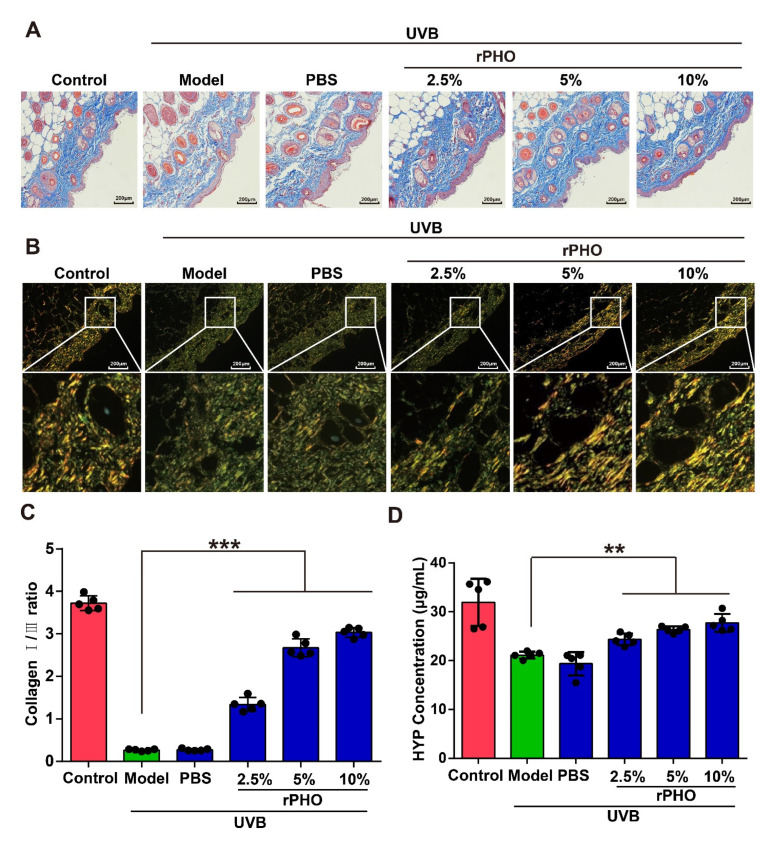
Photolyase inhibited UVB-induced reduction in collagen. (**A**) Masson staining; blue represents collagen fibers. (**B**) Representative picrosirius red staining of the dorsal skin. Images were obtained by polarized light microscopy. Type-I collagen: strong orange-yellow or bright red; type-III collagen: green. Scale bar = 200 μm. (**C**) The percentages of type-I and type-III collagen were calculated using ImageJ. (**D**) Hydroxyproline (HYP) in the dorsal skin was quantified. Data are shown as means ± SD (*n* = 5). ** *p* < 0.01, *** *p* < 0.001.

**Figure 6 antioxidants-11-02312-f006:**
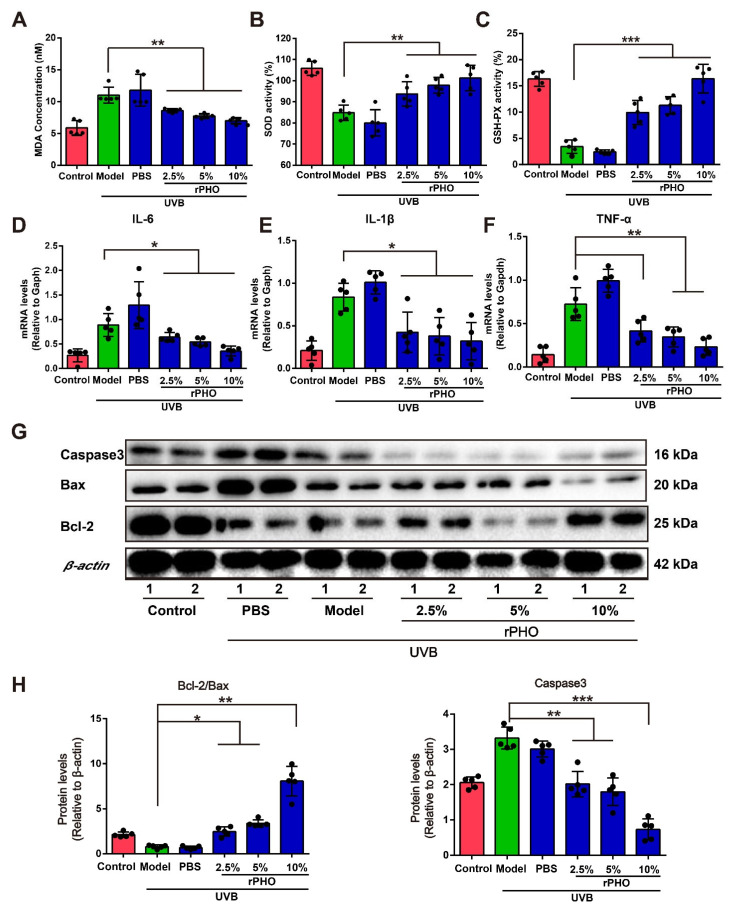
Photolyase significantly inhibited UVB-induced oxidative stress and inflammatory responses in mice. (**A**) Lipid peroxidation in the dorsal skin in each group after rPHO treatment was measured by quantifying malondialdehyde (MDA). (**B**) SOD activity in the dorsal skin after rPHO treatment. (**C**) GSH-Px activity in the dorsal skin after rPHO treatment. (**D**–**F**) Expression of IL-6, IL-1β, and TNF-α in skin was measured. mRNA relative levels were normalized to *Gapdh* levels. (**G**) Expression of caspase 3, Bcl-2, and Bax protein examined by western blotting. (**H**) Semi-quantification analysis of (**G**) by ImageJ. Data are expressed as means ± SD (*n* = 5). * *p* < 0.05, ** *p* < 0.01, *** *p* < 0.001.

**Figure 7 antioxidants-11-02312-f007:**
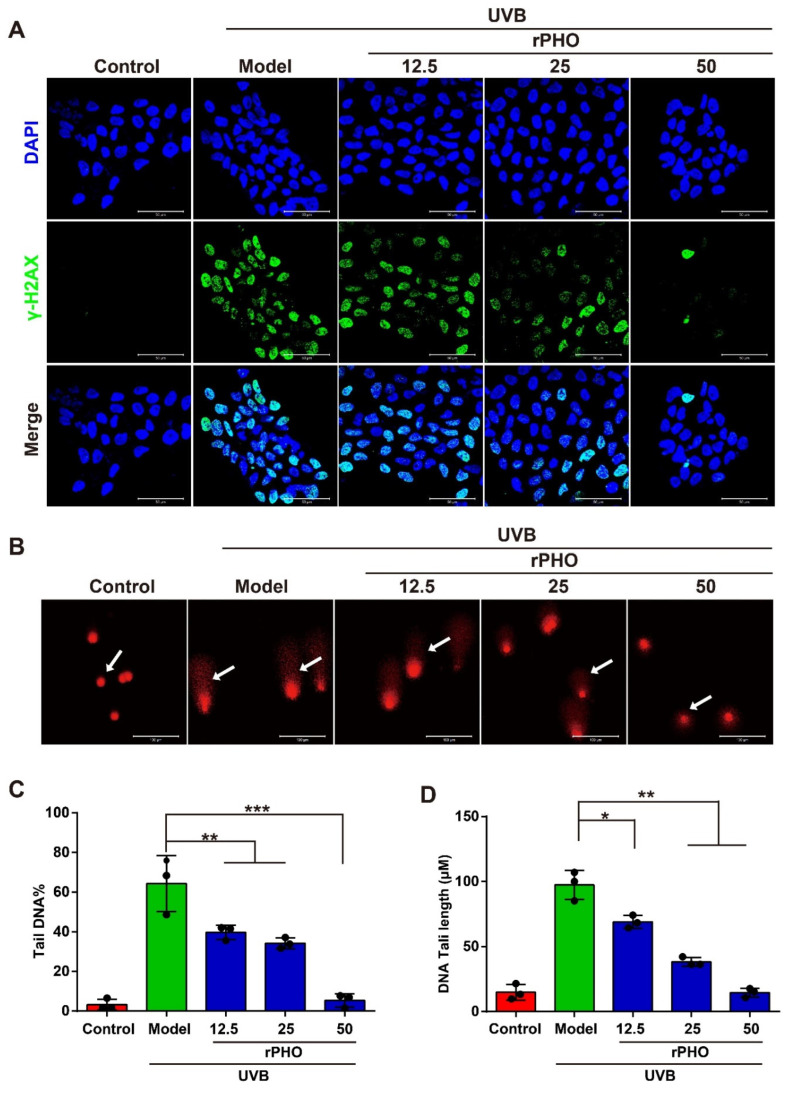
rPHO reduced UVB-induced DNA damage in HaCaT cells. (**A**) DNA damage measured by H2AX phosphorylation (γ-H2AX). HaCaT cells were treated with rPHO for 24 h after UVB irradiation (2 mJ/cm2), and immunostaining of γ-H2AX in HaCaT cells was performed. Blue: DAPI; green: γ-H2AX. (**B**) Representative alkaline comet assay images. Scale bar = 50 μm. (**C**,**D**) Tail DNA and DNA tail length were evaluated using ImageJ. Data are expressed as means ± SD (*n* = 3). * *p* < 0.05, ** *p* <0.01, *** *p* < 0.001.

**Figure 8 antioxidants-11-02312-f008:**
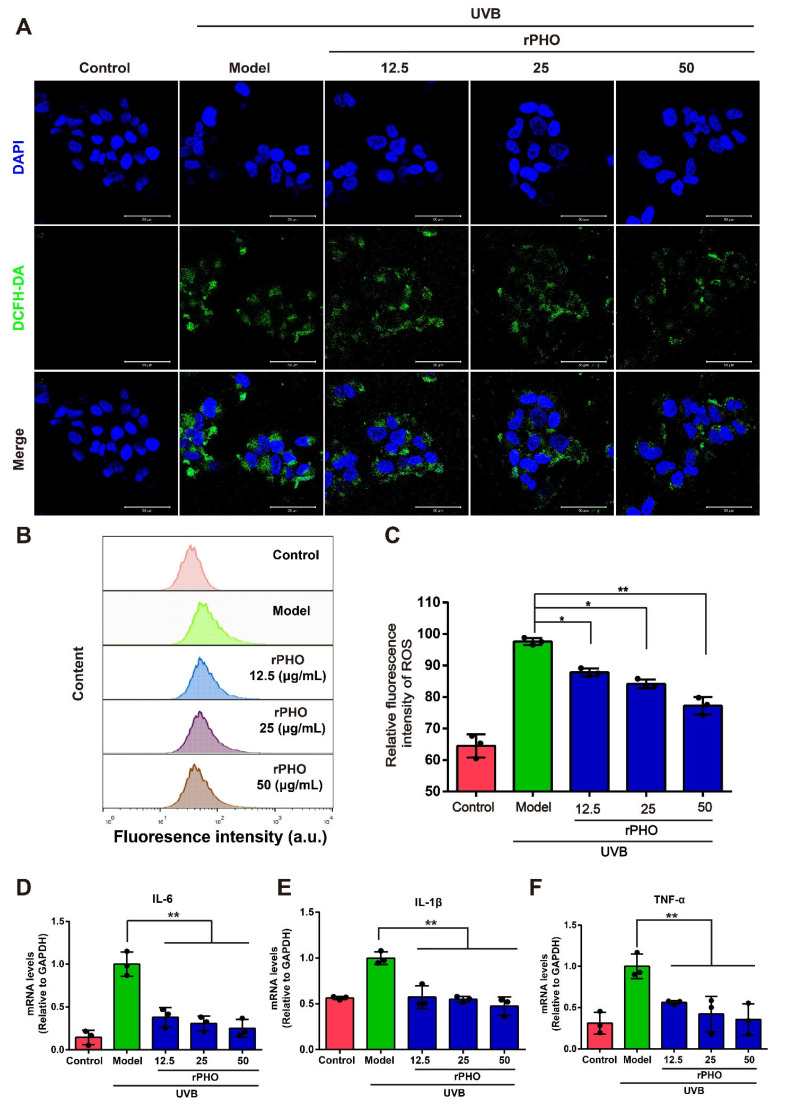
Photolyase inhibited UVB-induced ROS production and the release of inflammatory factors. (**A**) ROS production was analyzed by DCFH-DA staining under a fluorescence microscope. Scale bar = 50 μm. (**B**) ROS production was measured by flow cytometry. (**C**) Quantification of intracellular ROS levels. (**D**–**F**) The expression levels of IL-6, IL-1β, and TNF-α in HaCaT cells were measured. Relative levels of mRNA were normalized against levels of GAPDH. Data are expressed as means ± SD (*n* = 3). * *p* < 0.05, ** *p* < 0.01.

**Figure 9 antioxidants-11-02312-f009:**
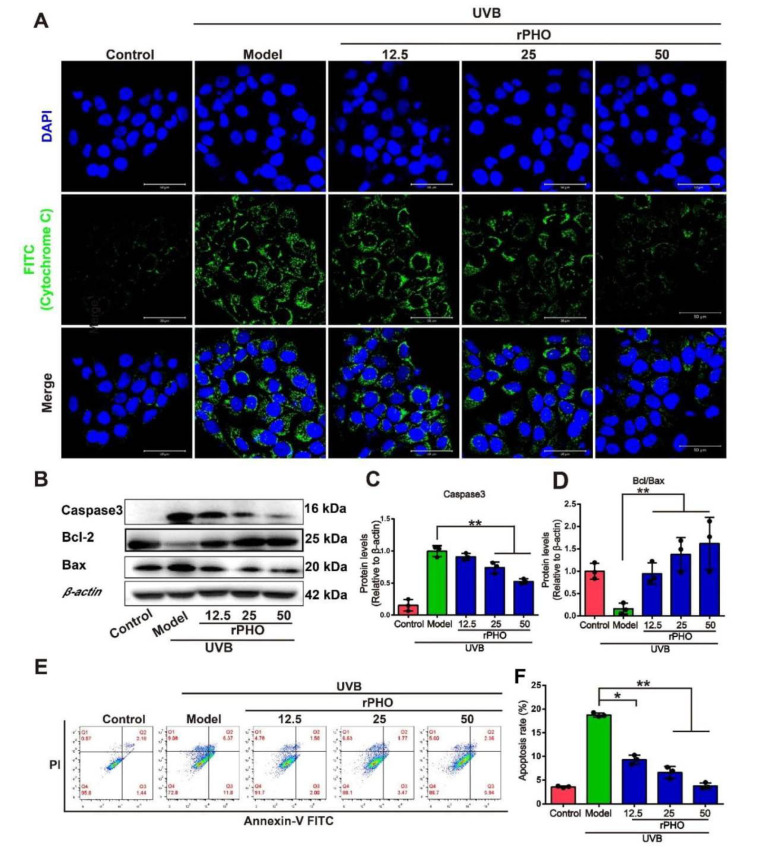
Photolyase inhibited UVB-induced apoptosis in HaCaT cells. (**A**) Release of cytochrome c in HaCaT cells treated with rPHO by immunofluorescence staining. Cytochrome c and nuclei are indicated by green and blue fluorescence, respectively. Scale bars = 100 μm. (**B**) Expression of caspase 3, Bcl-2, and Bax in HaCaT cells examined by western blotting. (**C**,**D**) Statistical analysis of protein expression in (**B**). (**E**) Apoptosis was measured by flow cytometry. (**F**) Statistical analysis of the apoptosis rates. Data are presented as means ± SD (*n* = 3). * *p* < 0.05, ** *p* < 0.01.

**Figure 10 antioxidants-11-02312-f010:**
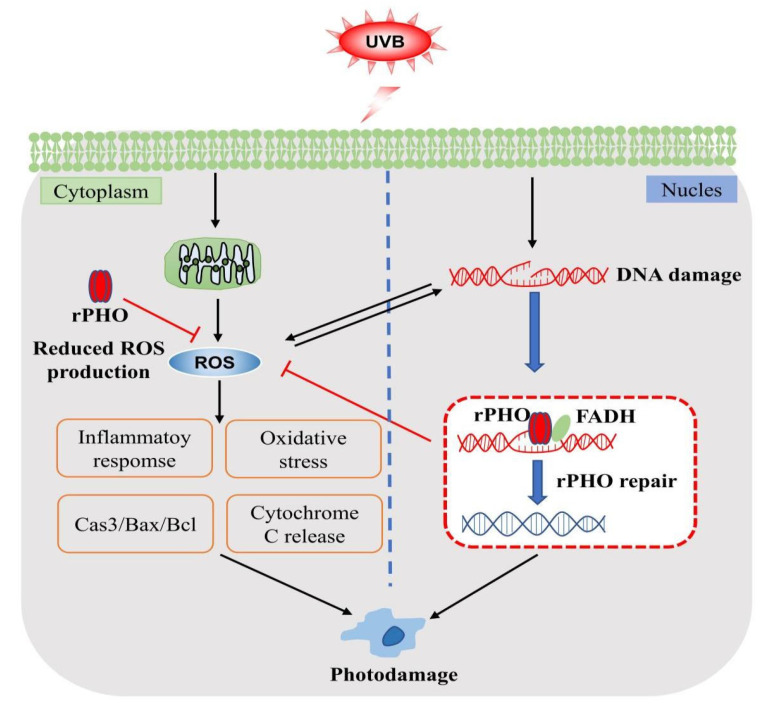
Mechanisms involved in protective effects of rPHO.

**Table 1 antioxidants-11-02312-t001:** Primers for qRT–PCR.

Gene Name	Sequence	Gene Accession Number
Mus-*Tnfα*-F	ATGTCTCAGCCTCTTCTCATTC	NM_001278601.1
Mus-*Tnf*α-R	GCTTGTCACTCGAATTTTGAGA	NM_001278601.1
Mus-*Il*-1β-F	GCCACCTTTTGACAGTGATGAG	NM_008361.4
Mus-*Il*-1β-R	GACAGCCCAGGTCAAAGGTT	NM_008361.4
Mus-*Il*-6-F	CACTTCACAAGTCGGAGGCT	NM_001314054.1
Mus-*Il*-6-R	CTGCAAGTGCATCATCGTTGT	NM_001314054.1
Mus-*Il*-10-F	GGAGGGGTTCTTCCTTGGGA	NM_010548.2
Mus-*Il*-10-R	TGAGCTGCTGCAGGAATGAT	NM_010548.2
Mus-*Gapdh*-F	TGTGTCCGTCGTGGATCTGA	NM_008084.4
Mus-*Gapdh*-R	CCTGCTTCACCACCTTCTTGA	NM_008084.4
Hum-TNF-a-F	TACGAGGAGGACGACTACCC	NM_000594.4
Hum-TNF-a-R	ATCCGGACACGGGTAAAACC	NM_000594.4
Hum-IL-B-F	GCACGAGTTCGGTAACCTCA	NM_000576.3
Hum-IL-B-R	ACTCCTTGACCGACACGAAC	NM_000576.3
Hum-IL-6-F	TCCGGTGGTGATGTTAACGG	NM_000600.5
Hum-IL-6-R	GCAGATATCGTGTGGGTGGA	NM_000600.5
Hum-GAPDH-F	CACCATCTTCCAGGAGCGAG	NM_001357943.2
Hum-GAPDH-R	AGAGGGGGCAGAGATGATGA	NM_001357943.2

## Data Availability

The data presented in this study are available in the article.

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
