# Peer review of "Recombinant Photolyase-Thymine Alleviated UVB-Induced Photodamage in Mice by Repairing CPD Photoproducts and Ameliorating Oxidative Stress"

_antioxidants, 2022, doi:10.3390/antiox11122312_

Round 1
Reviewer 1 Report
The submitted manuscript by Wang et al. deals with the use of a purified recombinant microbial photolyse as a protective agent against the deleterious effects of sunlight-induced DNA damage. The authors produced a recombinant photolyase that displays repair activity on UV-induced cyclobutane-pyrimidine dimers. This photolyase is apparently able to protect cells from UV damage both in vitro and in vivo in a mouse modell. The findings are promising and may constitute the basis for the development of photo-protective skin preparations.
The function of photolyases is to repair UV damage induced in the DNA molecule. To be plausible, the manuscript should be complemented with experimental evidence that the recombinant photolyase is taken up into cells and localizes to the nuclear compartment and/or mitochondria (both in cell cultures and in the mouse skin). How does the enzyme transfer through biological barriers like cell membranes? In the absence of a proof for an intracellular localization of the purified enzyme (once applied to cell cultures or animal skin), the results would become highly questionable.
Author Response
The function of photolyases is to repair UV damage induced in the DNA molecule. To be plausible, the manuscript should be complemented with experimental evidence that the recombinant photolyase is taken up into cells and localizes to the nuclear compartment and/or mitochondria (both in cell cultures and in the mouse skin). How does the enzyme transfer through biological barriers like cell membranes? In the absence of a proof for an intracellular localization of the purified enzyme (once applied to cell cultures or animal skin), the results would become highly questionable.
Response : Thank you for your valuable review and suggestions about our manuscript. We fully agree with the reviewer that photolyase uptake capacity in cell and mouse skin should be measured. Analysis of rPHO uptake capacity and skin permeability is an important first step before studying the function of rPHO in repairing UV damage.
In the revised manuscript, the fluorescein isothiocyanate (FITC)-labeled rPHO were used to examined the ability of HaCat to uptake rPHO. We found that incubation of HaCaT cells with FITC-labeled rPHO for 4 h result in robust uptake responses of FITC-labeled rPHO. And we observed green fluorescence in cytoplasm and nucleus, which indicated that the rPHO could be taken up into cells and localized to the nuclear and cytoplasm.
Additionally, to visualize the rPHO penetration pathway in the mouse skin, FITC-labeled rPHO carbomer gels were applied to the dorsal skin for 24 h. And the fluorescence was observed in the keratinocytes of the skin epidermis. Skin permeation of rPHO across mouse skin was presented in Figure 2.
As we know, transdermal delivery of protein is always challenged by its poor transdermal permeability. Transdermal drug delivery systems include transdermal patches, gels, and emulgels. In this study, to enhance the transdermal permeation of rPHO, we prepared rPHO carbomer gel. Presence of the carbomer resulted in ensuring optimal hydration for skin, then promote effective absorption of rPHO. Furthermore, photodamage often involves impaired skin barrier function, which makes rPHO more easily pass through the skin barrier. These factors contribute to enhance the transdermal permeation of rPHO. However, it is worth noting that transdermal absorption of rPHO may be limited. We warrant further exploration for transdermal and topical formulation application of rPHO. In revised manuscript, we discussed this issue in the Discussion section. Please check line 449-466.
“Additionally, transdermal delivery of protein is always challenged by its poor transdermal permeability. The rPHO uptake capacity and skin permeability is an im-portant first step before studying the function of rPHO in repairing UV damage. Paola Hernandez et al. reported a recombinant CPD-photolyase PhrAHym from the Antarctic bacterium Hymenobacter sp. UV11. Ant the PhrAHym repairs in a highly efficient way the CPD-photoproducts and reduces the γH2AX formation. These results are similar with those of our study[43]. However, the cells transfection with PhrAHym was achieved using Lipofectamine LTX (Invitrogen) transfection mix. The transfection mix cannot be used in reality applications. In our study, we demonstrated that rPHO could be uptake by HaCaT cells and keratinocytes.
Since transdermal drug delivery systems include transdermal patches, gels, and emulgels. In this study, to enhance the transdermal permeation of rPHO, we prepared rPHO carbomer gel. Presence of the carbomer results in ensuring optimal hydration for skin[44], then promote effective absorption of rPHO. Furthermore, photodamage often involves impaired skin barrier function[45], which makes rPHO more easily pass through the skin barrier. These factors contribute to enhance the transdermal permea-tion of rPHO. However, it is worth noting that transdermal absorption of rPHO may be limited. We warrant further exploration for transdermal and topical formulation ap-plication of rPHO.”
Reviewer 2 Report
The authors manufactured rPHO from Thermus thermophilus and verified its potential protection against UVB exposure.
The authors manufactured rPHO from Thermus thermophilus and verified its potential protection against UVB exposure. Additionally, additional information on the mechanism for rPHO should also be presented in detailed in the introductory section.
The methodology should describe how to remove endotoxin derived from Thermus thermophilus during rPHO purification.
Experimental conditions, equipment, and treatment conditions for UVB irradiation should be described in detail in the methodology.
A detailed and comprehensive discussion with reference to reference should be prepared rather than a consequential simple discussion of the inhibitory efficacy of UVB-induced phtotoxicity on rPHO.
Recently, a study related to photorepair in UVC for photolyase has been reported. (Environmental Toxicology and Pharmacology Volume 96, November 2022, 104001). In order to suggest the novelty of rPHO function, it would be good to present the result of comparison with photolyase.
Author Response
Point 1: The authors manufactured rPHO from Thermus thermophilus and verified its potential protection against UVB exposure. Additionally, additional information on the mechanism for rPHO should also be presented in detailed in the introductory section.
Response: Thank you for your constructive comments and suggestions. In revised manuscript, the mechanism for rPHO was added to the introductory section. Please check line 66-74.
“The rPHO belongs to the DNA photolyase class-1 family, which involves in repair of UV radiation-induced DNA damage and catalyzes the light-dependent monomerization (300-600 nm) of CPDs, which are formed between adjacent bases on the same DNA strand upon exposure to ultraviolet radiation. The amino acid sequence of rPHO have two different protein domains: (i) an antenna domain that holds the chromophore, and which transfers energy by resonance to FAD, and (ii) the photolyase domain that binds the FAD (responsible for removing the UV-induced DNA lesion). The FAD is non-covalently attached at the active site. After light absorbtion, rPHO repairs the pyrimi-dine dimers to their monomeric forms by transferring electrons to the dimers.”
Point2: The methodology should describe how to remove endotoxin derived from Thermus thermophilus during rPHO purification.
Response: We are sorry for the missing information. We have added the methodology of removing endotoxin to the method materials section. Please check line 91-103.
“Then the endotoxin removal agarose resin kit (YEASEN, 20518ES10) was used to remove endotoxin according to the manufacturer manual. In brief, the Endotoxin Removal Agarose Resin was fully mixed, and appropriate amount of grout was absorbed and added to the appropriate chromatographic column with the pyrogenless gun head, and bubbles were avoided. The lower outlet was opened to remove the protective liquid, and 3ml regenerated liquid was used for cleaning. The flow rate was controlled at 0.25ml/min, or < 10 drops /min. Repeat at least 2 times to ensure there are no endotoxins in the column. Then, 3 ml of regenerated liquid was used for cleaning and the flow rate was controlled at 0.25ml/min to remove endotoxins. The sample was added to the bal-anced resin, and the flow rate was adjusted at 0.25ml/min, or < 10 drops /min. When the outflow liquid drained out about 1ml, the outflow liquid was collected, and 1 ml of the equilibrium liquid was added to continue the collection.”
Point3: Experimental conditions, equipment, and treatment conditions for UVB irradiation should be described in detail in the methodology.
Response: Experimental conditions, equipment, and treatment conditions for UVB irradiation have been added to the method materials section. Please check line 106, line 133, and line 181.
Point4: A detailed and comprehensive discussion with reference to reference should be prepared rather than a consequential simple discussion of the inhibitory efficacy of UVB-induced phtotoxicity on rPHO.
Response: Thanks for your constructive suggestion. And inspired by the reviewer, we believed that the rPHO uptake capacity in cell and mouse skin was indeed a valid concern. Analysis of rPHO uptake capacity and skin permeability is an important first step before studying the function of rPHO in repairing UV damage. Therefore, in the revised manuscript, rPHO uptake capacity in cell and mouse skin were examined. The rPHO could be taken up into cells and localizes to the nuclear and cytoplasm in vitro. In animal experiments, the rPHO could be absorbed by the keratinocytes of the skin epidermis. Please check figure 2.
And in revised manuscript, we discuss this issue in the Discussion section. Please check line 449-466.
“Additionally, transdermal delivery of protein is always challenged by its poor trans-dermal permeability. The rPHO uptake capacity and skin permeability is an important first step before studying the function of rPHO in repairing UV damage. Paola Her-nandez et al. reported a recombinant CPD-photolyase PhrAHym from the Antarctic bacterium Hymenobacter sp. UV11. Ant the PhrAHym repairs in a highly efficient way the CPD-photoproducts and reduces the γH2AX formation. These results are similar with those of our study [43]. However, the cells transfection with PhrAHym was achieved using Lipofectamine LTX (Invitrogen) transfection mix. The transfection mix cannot be used in reality applications. In our study, we demonstrated that rPHO could be uptake by HaCaT cells and keratinocytes.
Since transdermal drug delivery systems include transdermal patches, gels, and emulgels. In this study, to enhance the transdermal permeation of rPHO, we prepared rPHO carbomer gel. Presence of the carbomer results in ensuring optimal hydration for skin [44], then promote effective absorption of rPHO. Furthermore, photodamage often involves impaired skin barrier function [45], which makes rPHO more easily pass through the skin barrier. These factors contribute to enhance the transdermal permea-tion of rPHO. However, it is worth noting that transdermal absorption of rPHO may be limited. We warrant further exploration for transdermal and topical formulation ap-plication of rPHO.”
Point5: Recently, a study related to photorepair in UVC for photolyase has been reported. (Environmental Toxicology and Pharmacology Volume 96, November 2022, 104001). In order to suggest the novelty of rPHO function, it would be good to present the result of comparison with photolyase.
Response: We thank the reviewer for mentioning this relevant literature. In the revised manuscript, we have discussed this study. The author reported a recombinant CPD-photolyase PhrAHym from the Antarctic bacterium Hymenobacter sp. UV11. Ant the PhrAHym repairs in a highly efficient way the CPD-photoproducts and reduces the γH2AX formation. These results are similar to those of our study. However, since the molecular weight of PhrAHym is 50.8 kDa, it is not readily taken up by living cells, the cell transfection with PhrAHym was achieved using Lipofectamine LTX (Invitrogen) transfection mix. The transfection mix cannot be used in reality applications. Therefore, the transdermal absorption of PhrAHym deserves further study.
Our study demonstrated that the rPHO uptake capacity in cell and mouse skin. However, it is worth noting that transdermal absorption of rPHO may be limited. We warrant further exploration for transdermal and topical formulation application of rPHO.
In addition, as the reviewer suggested that it would be good to present the result of comparison with photolyase. Unfortunately, however, there was no standard available for photolyase. We hope to address this issue in the future. And in the revised manuscript, we discuss this issue. Please check line 445-448.
“However, it would be good to present the result of comparison with photolyase. Unfortunately, there was no standard available for photolyase. Therefore, further studies are needed to confirm their precise function in photodamage by the standard of photolyase.”
Reviewer 3 Report
Manuscript can be accepted for publication after a major text revision.
- Introduction:
“…they contain the catalytic cofactor flavin adenine dinucleotide (FAD)…”
Do you mean FADH-?
- In the materials and methods part:
2.1. part:
Please expand the protein purification protocol and make it reproducible. Does your rPHO purification protocol consist of 1 stage (one Ni-NTA column)? Does rPHO contain His-tag?
Please describe at what moment have you added the cofactors to rPHO (was it just FADH or some other cofactor?).
2.2. part:
…The mixtures were incubated for 60 min under a daylight lamp. T…
Please include the characteristics of “a daylight lamp”.
- In the results part:
Mice were exposed to repetitive UVB (120 mJ/cm2) radiation for 15 days. Next, PBS or rPHO at concentrations of 2.5%, 5% and 10% were applied to the dorsal skin for 15 days before irradiation
Please clarify the sequence of events: did you apply rPHO before or after UVB irradiation?
“After 15 days of treatment, we observed erythema and wrinkles on the dorsal skin in UVB-irradiated (model group) and PBS-treated mice.
Why do PBS-treated mice exhibit erythema and wrinkles?
“The erythema and wrinkle formation were attenuated by rPHO treatment at 2.5%, 5% and 10%.”
Do you mean that 2.5% solution is 2,5 g of rPHO per 100g of gel?
- Discussion part:
“Though the photolyase gene from T. thermophilus has been cloned and sequenced [40, 41], little is known about its clinical value for skin photodamage and the molecular mechanisms underlying DNA damage repair after UVB irradiation.”
Relation of gene cloning and sequencing to its clinical value and mechanisms is unclear. Please clarify the sentence.
“Additionally, our results indicated that rPHO treatment can significantly attenuate the UVB-induced increases in the expression levels of TNF-α, IL-6, and IL-β in HaCaT cells. “
… significantly attenuate … increases …
Confusing sentence, please reformulate.
“…inhibited UVB-induced oxidative stress, and eliminated ROS production…”
Does your experiment show that rPHO eliminates ROS production after UVB-irradiation?
- Conclusion part:
“Rrecombinant photolyase-thymine protein…”
Double “r”
Suggestion for the conclusion part of the article: a couple of sentences with some speculation about the ways of rPHO transport to nucleus and possible rPHO interfering with NER system of mice will benefit the text.
Author Response
- Introduction:
“…they contain the catalytic cofactor flavin adenine dinucleotide (FAD)…”
Do you mean FADH-?
Response: We appreciate the reviewer’s meticulous attention to detail. FAD should be FADH-. We have corrected it in the revised manuscript. Please check line 51.
- In the materials and methods part:
2.1. part:
Please expand the protein purification protocol and make it reproducible. Does your rPHO purification protocol consist of 1 stage (one Ni-NTA column)? Does rPHO contain His-tag?
Response: The rPHO contains His-tag. Firstly, Ni-NTA column was used to purify the protein. Then the endotoxin removal agarose resin kit (YEASEN, 20518ES10) was used to remove the endotoxin. Detailed information concerning protein purification has been added to the method materials section. Please check line 85-103.
Please describe at what moment have you added the cofactors to rPHO (was it just FADH or some other cofactor?).
Response: The photolyase-thymine protein (rPHO) from Thermus thermophilus belongs to the DNA photolyase class-1 family. It involves in repair of UV radiation-induced DNA damage and catalyzes the light-dependent monomerization (300-600 nm) of cyclobutyl pyrimidine dimers (CPDs), which are formed between adjacent bases on the same DNA strand upon exposure to ultraviolet radiation. And the amino acid sequence of rPHO has two different protein domains: (i) an antenna domain that holds the chromophore, and which transfers energy by resonance to FAD, and (ii) the photolyase domain that binds the FAD (responsible for removing the UV-induced DNA lesion). The catalytic reaction of rPHO relies on the FADH− excitation by blue light. rPHO was responsible for the repair of carcinogenic DNA damage caused by ultraviolet radiation. They harbor the catalytic cofactor flavin adenine dinucleotide (FAD). The light-driven electron transfer from the excited state of the fully-reduced form of FADH− to the DNA lesions causes rearrangement of the covalent bonds, leading to the restoration of intact nucleobases.
Additionally, living tissues are poorly permeable to blue light. The rPHO utilizes secondary auxiliary chromophores that absorb blue light much more strongly than FADH−. The excitation energy of the secondary chromophores is transferred to the ground state of FADH−, leading to enhanced DNA repair by rPHO. As the secondary chromophores harvest photon energy for the catalytic reaction, they are called as light-harvesting chromophores or antenna chromophores.
Therefore, there is no need to add FADH− and cofactor during the catalytic process.
Nonetheless, adding exogenous FADH− and cofactor may boost the efficiency of the reaction. Inspired by the comments of the reviewer, we believe that it deserves more thorough investigation in future work.
To make the mechanism for rPHO easier to understand by readers, the mechanism for rPHO was added to the introductory section. Please check line 66-74.
2.2. part:
…The mixtures were incubated for 60 min under a daylight lamp. T…
Please include the characteristics of “a daylight lamp”.
Response: The power of the daylight lamp is 8W, and we have added detailed information in the revised manuscript. Please check line 109.
- In the results part:
Mice were exposed to repetitive UVB (120 mJ/cm2) radiation for 15 days. Next, PBS or rPHO at concentrations of 2.5%, 5% and 10% were applied to the dorsal skin for 15 days before irradiation
Please clarify the sequence of events: did you apply rPHO before or after UVB irradiation?
Response: We are very sorry to make you confused due to the unclear description in the original manuscript. We applied rPHO before UVB irradiation during the administration period.
And we have clarified the sequence of events in the revised manuscript. Please check line 180-186.
“The mice were placed in a relatively airtight box with plenty of food and water, and then an ultraviolet lamp (Philipp TUV8W) was placed on the upper end of the box and irradiated for 2h every day for 15 days, the blank group was not subjected to irradiation. Then the photolyase gel formulations of different concentrations were applied to the back of the mice. The PBS group was treated with gel formulations without rPHO. The dose was administered continuously for 15 days, and UVB irradiation continued for 2 hours daily during the duration of the dose (irradiation prior to application of the drug).”
“After 15 days of treatment, we observed erythema and wrinkles on the dorsal skin in UVB-irradiated (model group) and PBS-treated mice.
Why do PBS-treated mice exhibit erythema and wrinkles?
Response : PBS is a solvent control group. The skin of the mice in the PBS group appears red spots and folds, indicating that PBS did not have the activity of anti -UVB, and excluded unexpected effects of solvent on the photodamage.
“The erythema and wrinkle formation were attenuated by rPHO treatment at 2.5%, 5% and 10%.”
Do you mean that 2.5% solution is 2,5 g of rPHO per 100g of gel?
Response : 2.5% rPHO: 2.5 g of rPHO was dissolved into 100 ml PBS, and then 1g of carbomer was added to it and stirred until it was dissolved to form gel.
And the preparation process of rPHO gel was added to methods and materials. Please check line 172-173.
- Discussion part:
“Though the photolyase gene from T. thermophilus has been cloned and sequenced [40, 41], little is known about its clinical value for skin photodamage and the molecular mechanisms underlying DNA damage repair after UVB irradiation.”
Relation of gene cloning and sequencing to its clinical value and mechanisms is unclear. Please clarify the sentence.
Response: We appreciate the reviewer’s meticulous attention to detail. It is actually a wrong statement. This sentence has been corrected. Please check line 435-436.
“Though the photolyase gene from T. thermophilus has been cloned and sequenced [41, 42], however, the function of recombinant photolyase protein and its clinical value and mechanisms for skin photodamage are unclear.”
“Additionally, our results indicated that rPHO treatment can significantly attenuate the UVB-induced increases in the expression levels of TNF-α, IL-6, and IL-β in HaCaT cells. “
… significantly attenuate … increases …
Confusing sentence, please reformulate.
Response: Thanks for your kind reminder. We originally wanted to state that UVB irradiation can cause the increase of TNF-α, IL-6, and IL-β in HaCaT cells, while rPHO treatment inhibited the expression of TNF-α, IL-6, and IL-β.
We have corrected it in the revised manuscript. Please check line 443.
“Our results indicated that rPHO treatment can significantly inhibite the expression of TNF-α, IL-6, and IL-β in HaCaT cells.”
Round 2
Reviewer 3 Report
Dear authors,
The text is significantly improved and I can recommend this manuscript for publication.